# Amiodarone Enhances Anticonvulsive Effect of Oxcarbazepine and Pregabalin in the Mouse Maximal Electroshock Model

**DOI:** 10.3390/ijms22031041

**Published:** 2021-01-21

**Authors:** Monika Banach, Monika Rudkowska, Agata Sumara, Kinga Borowicz-Reutt

**Affiliations:** 1Independent Unit of Experimental Neuropathophysiology, Medical University of Lublin, Jaczewskiego 8b, PL-20-090 Lublin, Poland; monikabanach@umlub.pl (M.B.); monikarudkowska@umlub.pl (M.R.); 2Department of Pathophysiology, Medical University of Lublin, Jaczewskiego 8b, PL-20-090 Lublin, Poland; agatasumara@umlub.pl

**Keywords:** amiodarone, antiepileptic drugs, drug interactions, maximal electroshock-induced seizures

## Abstract

Accumulating experimental studies show that antiarrhythmic and antiepileptic drugs share some molecular mechanisms of action and can interact with each other. In this study, the influence of amiodarone (a class III antiarrhythmic drug) on the antiseizure action of four second-generation antiepileptic drugs was evaluated in the maximal electroshock model in mice. Amiodarone, although ineffective in the electroconvulsive threshold test, significantly potentiated the antielectroshock activity of oxcarbazepine and pregabalin. Amiodarone, given alone or in combination with oxcarbazepine, lamotrigine, or topiramate, significantly disturbed long-term memory in the passive-avoidance task in mice. Brain concentrations of antiepileptic drugs were not affected by amiodarone. However, the brain concentration of amiodarone was significantly elevated by oxcarbazepine, topiramate, and pregabalin. Additionally, oxcarbazepine and pregabalin elevated the brain concentration of desethylamiodarone, the main metabolite of amiodarone. In conclusion, potentially beneficial action of amiodarone in epilepsy patients seems to be limited by neurotoxic effects of amiodarone. Although results of this study should still be confirmed in chronic protocols of treatment, special precautions are recommended in clinical conditions. Coadministration of amiodarone, even at low therapeutic doses, with antiepileptic drugs should be carefully monitored to exclude undesired effects related to accumulation of the antiarrhythmic drug and its main metabolite, desethylamiodarone.

## 1. Introduction

Overlapping mechanisms of action between drugs provide a wide possibility of drug interactions. Both antiarrhythmic and antiepileptic drugs (AEDs) affect sodium, potassium, or calcium channels located in the brain and/or heart tissues.

Accumulating experimental studies show that some antiarrhythmic agents influenced, in fact, the threshold for electrically induced seizures and the action of antiepileptic drugs in the maximal electroshock test (MES) in mice. For instance, propafenone (a class Ic antiarrhythmic drug) at the dose range of 60–90 mg/kg significantly elevated the electroconvulsive threshold. Moreover, at the subthreshold dose of 50 mg/kg, propafenone potentiated the antielectroshock action of several antiepileptic drugs: valproate (VPA), carbamazepine (CBZ), phenytoin (PHT), phenobarbital (PB), oxcarbazepine (OXC), topiramate (TPM), and pregabalin (PGB) [1,2]. Mexiletine, a representative of the class Ib antiarrhythmic drugs, exhibited the characteristics of an antiseizure drug in the mouse model of electroconvulsions. Possible in such a case, isobolographic analysis revealed additive interaction between mexiletine and carbamazepine, phenytoin, and phenobarbital, respectively. In contrast, coadministration of mexiletine and valproate resulted in antagonistic interaction in the maximal electroshock test in mice [3]. Propranolol, metoprolol, verapamil, and diltiazem were the most widely investigated drugs among beta receptor and calcium channel blockers, that is, class II and IV antiarrhythmics, respectively. In the majority of cases, they showed antiseizure action in different animal seizure models, however the results of experiments were not unequivocal. Moreover, some antiarrhythmic drugs, especially when overdosed, presented proconvulsive effect [4].

Amiodarone, as a class III antiarrhythmic drug, is commonly used in the treatment of supraventricular and ventricular arrhythmias. Additionally, administration of amiodarone is recommended for cardiopulmonary resuscitation [5,6]. As a lipid-soluble multichannel blocker that can penetrate to the central nervous system, amiodarone seems to be a very interesting molecule. Its pharmacological profile involves inhibition of sodium and calcium L-type channels and the modulation of potassium outward current and antagonistic interaction with adrenergic receptors in the heart [7,8,9]. Additionally, in the brain, amiodarone was shown to decrease depolarization-evoked calcium-dependent glutamate release from the rat hippocampal synaptosomes [9] and elevate concentrations of inhibitory neurotransmitters in the rat medulla oblongata [10].

Despite documented molecular mechanism of action that may justify the anticonvulsive and neuroprotective effect of amiodarone, experimental data in this area are scarce and ambiguous. In the pentylenetetrazole- and caffeine-induced seizures, amiodarone at doses of 100–150 mg/kg prolonged both latency period and time to death. Additionally, in the former seizure model, the convulsion rate and mortality were slightly reduced [11]. No significant impact of amiodarone (40 mg/kg) on seizures or mortality was documented in the mouse model of cocaine toxicity [12]. In another study, amiodarone was administered at doses of 50–150 mg/kg after transient forebrain ischemia in rats. No deterioration in the spatial cognitive function or neuronal survival in the hippocampal CA1 region was observed [13]. Amiodarone pretreatment reduced the extent of brain ischemic insult, decreasing infarct volume and improving neurological outcomes in mice [14]. In contrast, in another murine model of hypoxic-ischemic brain injury, amiodarone worsened survival rates and neurological outcomes. This effect was associated with intracellular accumulation of sodium and subsequent brain edema due to impaired transmembrane sodium transportation [15].

Previously, we demonstrated that amiodarone at doses of 25–75 mg/kg did not affect the electroconvulsive threshold in mice. Additionally, administered at the dose of 75 mg/kg, amiodarone significantly enhanced the antielectroshock action of carbamazepine in the maximal electroshock test in mice [16]. Interesting results of this study prompted us to extend the research with four novel antiepileptic drugs: oxcarbazepine, pregabalin, lamotrigine (LTG), and topiramate in the same seizure model.

The choice of antiepileptic drugs was mainly dictated by their diverse mechanisms of action. Lamotrigine binds to slowly inactivating conformation of voltage-gated sodium channels (VDSCs), limiting sustained repetitive firing of neurons, reduces synaptic release of excitatory amino acids and may also inhibit dendritic action potentials and enhance currents that are carried by hyperpolarization-activated cation channels (h channels) [17]. The way of action of oxcarbazepine involves a blockade of sodium currents and modulation of calcium channels [18]. Topiramate activates GABA A receptors, blocks glutamatergic AMPA/kainite receptors, inhibits type L calcium channels and sodium currents, activates potassium channels, and reduces carbonic anhydrase activity [17]. Pregabalin reduces the synaptic release of several neurotransmitters by binding to the alpha_2_-delta protein, an auxiliary subunit of voltage-gated calcium channels [19]. Such a differentiated mechanisms of action of antiepileptics increases the chances to understand the background of possible pharmacodynamic interactions with amiodarone. Importantly, in experimental conditions, all selected antiepileptic drugs displayed antiseizure activity in the maximal electroshock test in mice [17].

Another argument for choosing second-generation antiepileptic drugs was their increasing use in medical practice. Pregabalin has been approved for adjunctive therapy of partial seizures in adults, whereas the remaining three antiepileptics are recommended for the treatment of focal and generalized epilepsy as a single drug or add-on medication. Other indications involve bipolar disorders (lamotrigine), prevention of migraine (topiramate), treatment of neuropathic pain (lamotrigine, oxcarbazepine, pregabalin). Growing prescription of the newer antiepileptic drugs both in epileptic and nonepileptic patients creates the possibility of coadministration with other drugs, such as amiodarone. Despite favorable pharmacokinetic characteristics and improved safety profile of the newer antiepileptics, there is still a risk of drug interactions. In our opinion, evaluation of possible interactions between amiodarone and selected antiepileptic drugs may be a valuable contribution to the current knowledge.

## 2. Results

### 2.1. Maximal Electroshock Test

In our previous article, we showed that amiodarone at doses of 25, 50, and 75 mg/kg did not influence the electroconvulsive threshold [16]. In this study, amiodarone at two doses of 75 and 100 mg/kg were tested in combination with antiepileptic drugs against the maximal electroshock test.

Amiodarone at the dose of 100 mg/kg significantly potentiated the anticonvulsive properties of oxcarbazepine, lowering its ED_50_ value from 11.9 ± 1.23 to 8.2 ± 0.77 mg/kg [F(2.33) = 6.593, *p* = 0.0019]. Furthermore, amiodarone at doses of 87.5 and 100 mg/kg increased the protective action of pregabalin in the maximal electroshock test in mice. The respective ED_50_ values were 65.6 ± 16.97 and 57.8 ± 11.58 mg/kg, whereas the control ED_50_ of pregabalin was 135.3 ± 15.21 mg/kg [F(3.132) = 10.47, *p* < 0.0001]. The action of the two remaining antiepileptic drugs, lamotrigine and topiramate, was not affected by amiodarone in the maximal electroshock test (see Figure 1A–D).

### 2.2. Chimney Test and Passive Avoidance Task

Pregabalin, at the dose of 57.8 mg/kg administered alone and in combination with amiodarone, significantly impaired motor coordination in mice (see Table 1). Amiodarone at the dose of 100 mg/kg administered alone and in combination with three antiepileptic drugs—lamotrigine, oxcarbazepine, and topiramate—did not disturb motor coordination in the chimney test.

Amiodarone at the dose of 100 mg/kg given alone and in combination with antiepileptic drugs, significantly impaired long-term memory in mice. Remarkably, all of the tested antiepileptic drugs did not influence the retention time in the passive-avoidance task in comparison to the control group (see Table 2).

### 2.3. Brain Concentrations of Antiepileptic Drugs and Amiodarone

Amiodarone did not affect the total brain concentration of the tested antiepileptic drugs (see Figure 2A–D). In contrast, oxcarbazepine, topiramate, and pregabalin significantly increased the total brain concentration of amiodarone. Interestingly, oxcarbazepine and pregabalin had the same effect on the concentration of DEA (see Figure 3).

Results are presented as the means ± SD (in µg/mL) of at least eight determinations. Statistical analysis was performed using the unpaired Student’s test. AMD—amiodarone, LTG—lamotrigine, OXC—oxcarbazepine, TPM—topiramate, and PGB—pregabalin.

## 3. Discussion

In the presented study, we investigated the effects of coadministration of the antiarrhythmic agent amiodarone on the anticonvulsive action of four novel antiepileptic drugs—lamotrigine, oxcarbazepine, topiramate, and pregabalin—in the maximal electroshock test in mice. Amiodarone significantly potentiated the protective activity of oxcarbazepine and pregabalin, but not that of lamotrigine or topiramate against electrically induced seizures in mice. Additionally, amiodarone, given alone or in combination with lamotrigine, oxcarbazepine, or topiramate, impaired long-term memory in the step-through passive avoidance task in mice. Pregabalin, the combination of pregabalin and amiodarone, but not amiodarone alone, significantly impaired motor coordination in the chimney test in mice. The total brain concentrations of antiepileptic drugs were not affected by amiodarone. However, oxcarbazepine, topiramate, and pregabalin significantly elevated the brain concentration of the antiarrhythmic agent. Additionally, both oxcarbazepine and pregabalin had the same impact on DEA, an active metabolite of amiodarone. Since the two mentioned antiepileptic drugs significantly elevated the brain concentration of amiodarone and DEA, the nature of interactions between amiodarone and oxcarbazepine or pregabalin should be considered at least in part as pharmacokinetic. Nevertheless, increased levels of amiodarone should not be responsible for enhanced antielectroshock properties of oxcarbazepine or pregabalin, because, administered alone, it did not affect the electroconvulsive threshold in mice [16]. Additionally, the influence of DEA on seizure phenomena seems to be unlikely, although it cannot be completely excluded. No literature data refer to this problem.

In our previous study, amiodarone administered at the dose of 75 mg/kg significantly enhanced the antielectroshock action of carbamazepine. No pharmacokinetic interactions between those two drugs were revealed, since the concentration of carbamazepine was not influenced by coadministration of amiodarone. Unfortunately, neither concentration of amiodarone nor its main metabolite was assessed in this study [16].

Oxcarbazepine, as a derivative of carbamazepine, is commonly believed to share the mechanism of action with the parent drug; primarily, the inhibition of sodium currents is taken into account. The main difference between the two antiepileptic drugs concerns calcium channels. Carbamazepine blocks mainly L-type, whereas oxcarbazepine (or rather its active monohydroxy derivative) N/P and R types of calcium currents [18]. Amiodarone enhanced the antiseizure action of both antiepileptic drugs; however, in the case of oxcarbazepine, a higher dose (100 mg/kg) of the antiarrhythmic drug was required. It seems that more advantageous interactions between amiodarone and carbamazepine may be due to their supplementary action on L-type calcium channels.

Pregabalin, the second drug the action of which was potentiated by amiodarone, does not share any mechanism of action with this antiarrhythmic drug. It binds to an auxiliary subunit of voltage-gated calcium channels (the alpha_2_-delta protein) and decreases synaptic release of neurotransmitters [19]. Amiodarone at the dose of 87.5 mg/kg lowered the ED_50_ value of pregabalin by 51%. It may be assumed that amiodarone potentiates the action of pregabalin by complementing its mechanism of action with amiodarone-dependent modulation of Na^+^, K^+^, and Ca^2+^ channels. Nevertheless, complementary action should lead to positive interactions also with lamotrigine and topiramate, but such an effect has not been observed. Therefore, further molecular and pharmacokinetic studies are necessary to explain the nature of the above interactions.

Among other class III antiarrhythmic drugs, dronedarone and sotalol present some similarities to the action of amiodarone, particularly regarding K^+^ channel blocking. Both drugs were evaluated in the electrically induced seizure tests in mice and surprisingly exhibited diverse effects. Dronedarone has been developed as a less toxic alternative to amiodarone. When applied at the dose of 75 and 100 mg/kg, it significantly elevated the electroconvulsive threshold in mice [20]. Furthermore, dronedarone at the dose of 50 mg/kg potentiated the antiseizure action of lamotrigine [21]. In contrast, dronedarone given at the same dose alleviated the protective action of phenytoin in the maximal electroshock test, increasing its ED_50_ value [20]. Despite some similarities in the structure between amiodarone and dronedarone, their interactions with antiepileptic drugs were completely different. Dronedarone did not affect brain concentrations of phenytoin or lamotrigine in mice, so revealed interactions seem to have pharmacodynamic nature. Regretfully, brain concentration of dronedarone was not assessed [20,21].

Sotalol at doses up to 100 mg/kg did not affect the electroconvulsive threshold in mice. In the maximal electroshock test in mice, sotalol enhanced protective action of valproate and phenytoin, whereas that of carbamazepine, phenobarbital [22], lamotrigine, oxcarbazepine, topiramate, and pregabalin [data not published] were not affected. No pharmacokinetic interactions were observed between sotalol and classical antiepileptic drugs. Remarkably, sotalol increased the brain concentration of topiramate and oxcarbazepine and lowered that of lamotrigine without affecting their antielectroshock activity.

Unfortunately, analysis of presented interactions between class III antiarrhythmic drugs and antiepileptic drugs does not allow us to draw any plausible predictions for future studies and clinical practice. No regularity or common effects of class III antiarrhythmic drugs on seizure phenomena were found in mice.

Additionally, pharmacokinetics of amiodarone further hamper the reliable inference. Metabolism and tissue distribution of this antiarrhythmic drug was described previously in more detail [16]. Several studies compared features of amiodarone and its metabolite DEA. Pharmacokinetic analysis indicates that a constant fraction (10%) of amiodarone is converted to DEA by cytochrome P450 oxidase (CYP3A4), and this process is independent on the dose of the parent drug. Amiodarone was reported to inhibit P-glycoprotein and hepatic enzymes (CYP1A2, CYP2C9, CYP2D6, CYP3A4), as well as to decrease elimination of several drugs [8,23,24,25]. Both CYP3A4 and P-glycoprotein are also inhibited by DEA [25]. The cytochrome P450 enzymes are not involved in metabolism of pregabalin nor lamotrigine, whereas oxcarbazepine and topiramate are believed to be weak inducers of CYP3A4 and inhibitors of CYP2C19. Antiepileptic drugs are also believed to be substrates of P-glycoprotein, an efflux transporter at the blood–brain barrier [26]. However, in our study, amiodarone did not influence the brain concentration of the novel antiepileptic drugs.

On the other hand, three of the novel antiepileptics (oxcarbazepine, topiramate, and pregabalin) increased the brain concentration of amiodarone in mice. Additionally, oxcarbazepine and pregabalin also elevated the brain concentration of DEA. Some other drugs, for example antiviral (asunaprevir, boceprevir) or antidepressant (fluoxetine, escitalopram) medications were reported to increase amiodarone concentration, but the mechanism of this effect is not clear. No data about the influence of the abovementioned drugs on DEA concentrations are available [27]. Since oxcarbazepine can stimulate CYP3A4, it may explain increased production of DEA, but not increased concentration of the parent drug. Moreover, hepatic enzyme induction may take several days before being established. We should remember that pharmacokinetic drug–drug interactions occur not only at the stage of drug metabolism. Drug absorption, tissue distribution, or elimination should also be considered [26]. Oxcarbazepine and pregabalin increased the brain concentration of the parent drug and its metabolite. One may speculate that preferential brain distribution or inhibition of elimination from the central nervous system occurs. However, it cannot be excluded that the pharmacokinetics of drugs used in this study varies greatly between humans and mice. Probably, measuring the plasma concentrations of amiodarone and DEA could help in further conclusions. To our knowledge, there is no scientific data dealing with the regulation of blood–brain barrier permeability for amiodarone by antiepileptic drugs.

The principal metabolite of amiodarone is less lipid soluble and, except for fat tissue, accumulates in tissues at higher concentrations than the parent molecule [25]. There are no data regarding the action of DEA against electrically induced seizures in mice. In the heart, DEA is believed to be a stronger inhibitor of inward sodium current and a weaker inhibitor of inward calcium current than amiodarone [28,29,30]. In in vitro conditions, Kodavanti et al. [31] observed that amiodarone and DEA increased Ca^2+^ influx from the external medium and produced prolonged elevation of Ca^2+^ concentration in rat brain synaptosomes. On the other hand, there is no simple correlation between amiodarone or DEA plasma concentrations and their clinical effects [25].

To our knowledge, the presented study showed for the first time that topiramate significantly elevates concentrations of amiodarone in mouse brains, while oxcarbazepine and pregabalin increases the brain concentrations of both amiodarone and DEA. This mechanism of this effect remains vague. It is tempting to speculate, however, that DEA is responsible for the beneficial impact of the parent molecule on the anticonvulsive action of oxcarbazepine and pregabalin.

Another interesting finding in this paper is that amiodarone at the dose of 100 mg/kg given alone or in combination with antiepileptic drugs disrupts long-term memory in the passive avoidance test in mice. In our previous study, the combined treatment with classical antiepileptic drugs and amiodarone at the dose of 75 mg/kg did not affect long-term memory. Amiodarone is known to evoke several serious adverse effects, especially during chronic therapy, but memory impairment is not mentioned in the current literature [5,32].

Curiously, in a recently published article, Kotoda et al. [33] showed that amiodarone exhibited a dose-dependent analgesic effect in three mouse models of pain—the acetic-acid-induced writhing test, formalin test, and tail withdrawal test. The antiarrhythmic drug applied at doses of 100–200 mg/kg reduced the nociceptive response in all tests in mice. Moreover, this effect was reversed by veratridine, an alkaloid preventing inactivation of voltage-gated sodium channels. Therefore, the analgesic effect of amiodarone is possibly due to a blockade of sodium channels. In light of these reports, the results obtained in the passive avoidance test should be probably reinterpreted. Since the perception of an aversive stimulus may be alleviated after amiodarone application, a current intensity used in the present study could be too weak to be remembered by animals. It is worth emphasizing that both effects of amiodarone—analgesic and disruption of long-term memory in mice—were dose-dependent. This fact may further support our theory explaining presented results of the passive avoidance test in mice. Nevertheless, additional behavioral studies are necessary to verify this assumption and to assess cognitive function in mice receiving amiodarone.

In the chimney test, pregabalin given alone at the dose of 57.8 mg/kg and in combination with amiodarone impaired motor coordination in mice. A similar effect of pregabalin, applied at comparable doses, was, however, reported in several previous studies employing the same behavioral test [2,21]. Additionally, dronedarone (50 mg/kg) coadministered with pregabalin at the dose of 195.5 mg/kg, but not alone, considerably impaired motor coordination evaluated in the chimney test in mice. Nevertheless, neurotoxic symptoms induced by drug combinations seem to be dependent rather on the action of pregabalin, not that of amiodarone or dronedarone [21].

Summing up, the present study demonstrated that amiodarone enhances the anticonvulsive potency of oxcarbazepine and pregabalin in the mouse model of tonic-clonic seizures. Remarkably, the antiarrhythmic agent given alone and in combination with antiepileptic drugs evoked significant long-term memory disruption, which seems to argue against coadministration of these drugs. More advanced neurochemical studies are necessary to explain pathogenesis of observed undesired effects. In the next step, results of this study should be confirmed in chronic protocols of treatment. Nevertheless, practical application of the obtained results seems to be limited by toxic effects of chronic amiodarone. In clinical conditions, even coadministration of amiodarone at lower doses with antiepileptic drugs should be carefully monitored to exclude undesired toxic effects that may result from elevated concentrations of the antiarrhythmic drug and its main metabolite.

## 4. Materials and Methods

### 4.1. Animals

All study experiments were carried out on adult female Swiss mice weighing 20–25 g. The animals were kept in colony cages under standardized laboratory conditions: a natural light-dark cycle 12/12 h, temperature 20–24 °C, air humidity 45–65%, air exchange 15/h, free access to tap water and food (chow pellets). After 7 days of acclimatization, the mice were randomly assigned to experimental groups (8–10 animals). All the investigations were approved by Local Ethical Committee at University of Life Sciences in Lublin (32, 19 Jun 2015 and 37, 5 Mar 2018) and complied EU Directive 2010/63/EU for animal experiments as well as ARRIVE guidelines.

### 4.2. Drugs

Amiodarone (Opacorden, Polpharma, Starogard Gdański, Poland), lamotrigine (Lamitrin, GlaxoSmithKline, Brentford, Middlesex, UK), oxcarbazepine (Trileptal, Novartis, Nürnberg, Germany), topiramate (Topamax, Janssen-Cilag International NV, Beerse, Belgium), pregabalin (Lyrica, Pfizer Limited, Sandwich, Kent, UK) were suspended in a 1% aqueous solution of Tween 80 (Sigma-Aldrich, St. Louis, MO, USA). All drugs or vehicle (in control groups) were administered intraperitoneally (*i.p.*) in a volume of 10 mL/kg body weight. Amiodarone and oxcarbazepine were applied 30 min., lamotrigine and topiramate 60 min., pregabalin 120 min. before tests, respectively. Doses of antiepileptic drugs and amiodarone, as well as their application time, were based on our experimental experiences published elsewhere [2,16].

### 4.3. Maximal Electroshock Seizure Test

The maximal electroshock seizure (MES) test is a well-known animal model of tonic-clonic seizures commonly used in the preclinical evaluation of anticonvulsive properties of the tested molecules [34].

Convulsions were evoked by an electric stimulus (an alternating current 25 mA, 50 Hz, 500 V, 0.2 s) generated by a rodent shocker (Hugo Sachs Elektronik, Freiburg, Germany) and delivered via ear-clip electrodes. Tonic hindlimb extension (i.e., hindlimbs of animals outstretched 180˚ to plane of the body axis) was established as the endpoint. ED_50_ is a median effective dose of the tested drug that protects 50% of mice against maximal electroshock-induced seizures. To calculate ED_50_ of the drug, at least 3 groups of animals received progressive doses of antiepileptic drugs alone or in combination with amiodarone and were challenged with the maximal electroshock test. A dose-response curve was calculated on the basis of the percentage of mice protected (protection in less than 50%, around 50%, and more than 50% of animals) according to Litchfield and Wilcoxon [35].

### 4.4. Chimney Test

The effects of amiodarone, antiepileptic drugs (at their ED_50s_ values), and combinations of amiodarone with antiepileptic drugs on motor coordination in mice were determined in the chimney test [36]. On the first day of the test, mice were placed individually in the plastic tube (3 cm inner diameter, 25 cm long) and allowed to go on. When the animal reached the end of the tube, the chimney was positioned vertically, and the mouse had to climb backward up within 60 s. The next day the same procedure was repeated after drug administration. Results are shown as a percentage of animals that failed to perform the test.

### 4.5. Step-Through Passive-Avoidance Test

The step-through passive-avoidance test is based on natural aversion of rodents to bright places and is regarded as a measure of long-term memory [37]. The procedure was carried out using a Multi Conditioning System (MCS, TSE Systems GmbH, Bad-Homburg, Germany), a computer-controlled apparatus automatically recording step-through latencies. It also enables entire isolation of animals from all external stimuli that may interfere with mouse behavior. The MCS software features are compliant with the Good Laboratory Practice.

The apparatus consists of two compartments with a stainless-steel bar floor, separated by a sliding door. On the first day of the experiment, the tested mouse after drug injection was placed in the light part of the apparatus and allowed to enter the dark chamber. Upon entering the dark compartment with all four paws, the sliding door was automatically closed, and a weak electrical current was delivered via the grid floor (2 s shock exposure duration, 0.3 mA). After completion of the training test, the mouse was returned to the home cage. Retention latencies were determined 24 h later (on the second day of the experiment). Each animal was placed into the light compartment with open access to the dark compartment and observed for 180 s. The entrance to the dark box within this time was recorded and considered as evidence of the long-term memory impairment. The results were presented as medians (with 25, 75 percentiles) of the time until the animals enter the dark box.

### 4.6. Measurement of Brain Concentrations of Antiepileptic Drugs and Amiodarone

Total brain concentrations of antiepileptic drugs and amiodarone were measured to detect possible pharmacokinetic interactions between those drugs.

Control animals were administered with one of the antiepileptic drugs and saline. The examined groups were administered with the respective antiepileptic drug and amiodarone at the dose of 100 mg/kg. According to the procedure, mice were killed by decapitation at times scheduled for the maximal electroshock test. Then, the brains were removed, weighed, and homogenized (Ultra Turax T8 homogenizer, IKA, Staufen, Germany) with Abbott buffer (2:1 vol/weight). Homogenates were centrifuged at 10,000× *g* for 10 min, and supernatants (75 μL) were analyzed for drug content.

Brain concentrations of antiepileptic drugs were determined by fluorescence polarization immunoassay, using an Architect c4000 clinical chemistry analyzer (Abbott Laboratories Poland). The results were expressed in μg/mL and subsequently computed as means ± SD of at least eight determinations.

To evaluate brain concentrations of amiodarone, liquid chromatography mass spectrometry was applied. Samples were mixed with cold 1:1 *v/v* methanol: ethanol at 1:3 *v/v* ratio, vortexed, placed at −20°C for 15 min, then vortexed again and centrifuged at 4 °C for 5 min. Supernatants were subjected to liquid chromatography mass spectrometry analysis (LC-MS). An Agilent Technologies liquid chromatograph 1290 Infinity series coupled to an Agilent Technologies quadrupole time-of-flight mass spectrometer 6550 iFunnel LC/QTOF equipped with a Jet Stream Technology ion source was employed. Separations were carried out using a Zorbax Extend C18 RRHT 2.1 × 100 mm 1.8 μm column and water and acetonitrile both with the addition of 0.1% *v/v* formic acid as mobile phases. HRMS spectra were acquired in the positive polarity at the range of 100 to 1000 m/z. Internal mass calibration was enabled, two reference ions of m/z 121.0509 and 922.0058 were used, to ensure mass measurement accuracy < 1 ppm. Agilent Technologies Mass Hunter software, B.10 for the acquisition and B.07 for the data processing, was utilized. For quantification of amiodarone [M+H]^+^ ions of m/z 646,0310 were extracted. EIC peaks were integrated, and peak areas, which are proportional to analyte concentrations, were reported. Additionally, amiodarone’s main metabolite—desethylamiodarone (DEA)—concentrations were also measured.

### 4.7. Statistics

The ED_50_ values with their respective 95% confidence limits were calculated in the computer log-probit analysis according to Litchfield and Wilcoxon [35]. Then, the standard errors (SEMs) of the mean values were assessed on the basis of confidence limits. Multiple comparisons of the ED_50_ values (±SEM) from the MES test were performed using one-way analysis of variance (ANOVA) followed by the post hoc Tukey test.

The Fisher’s exact probability test was used to analyze qualitative variables from the chimney test. The non-parametric Kruskal–Wallis test was used for statistical assessment of results obtained in the passive-avoidance test.

Brain concentrations of antiepileptic drugs were evaluated by the use of the unpaired Student’s *t* test, whereas concentrations of amiodarone and its metabolite were statistically verified using one-way analysis of variance (ANOVA) followed by the post hoc Dunnett’s test. The significance level was set at *p* ≤ 0.05.

## Figures and Tables

**Figure 1 ijms-22-01041-f001:**
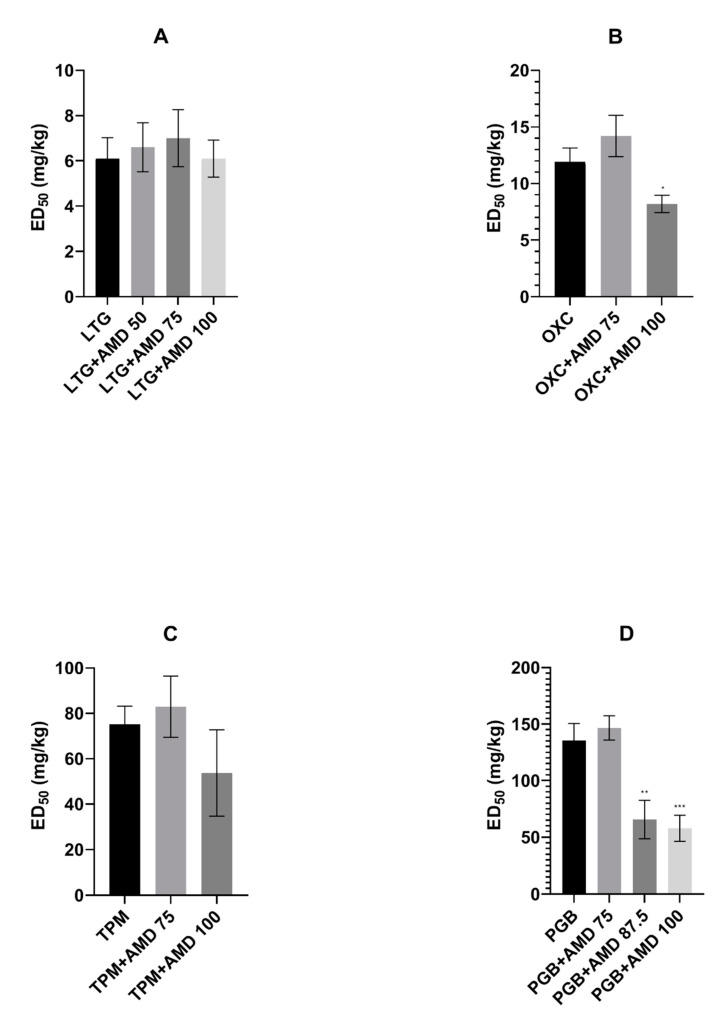
Effect of amiodarone (AMD) on the anticonvulsant action of (**A**). lamotrigine (LTG), (**B**). oxcarbazepine (OXC), (**C**). topiramate (TPM), and (**D**). pregabalin (PGB) against maximal electroshock-induced seizures in mice. Data are presented as median effective doses (ED_50_ in mg/kg) with standard errors (SEM) that protect 50% of animals from the tonic-clonic seizures. Statistical analysis of data was performed using one-way analysis of variance (ANOVA) followed by the post hoc Tukey test. * *p* < 0.05, ** *p* < 0.01, *** *p* < 0.001 vs. control group.

**Figure 2 ijms-22-01041-f002:**
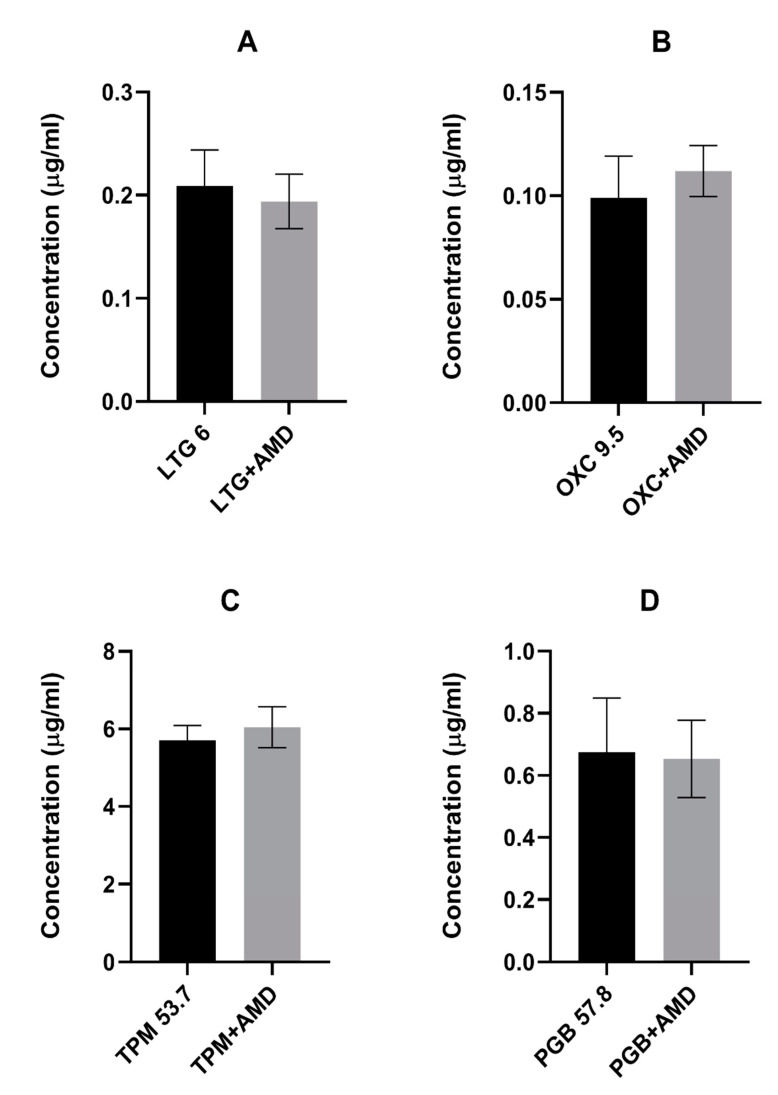
Effects of acute amiodarone on the brain concentrations of: (**A**). lamotrigine, (**B**). oxcarbazepine, (**C**). topiramate and (**D**). pregabalin. Results are presented as the means ± SD (in µg/mL) of at least eight determinations. Statistical analysis was performed using the unpaired Student’s test. AMD—amiodarone, LTG—lamotrigine, OXC—oxcarbazepine, TPM—topiramate, PGB—pregabalin.

**Figure 3 ijms-22-01041-f003:**
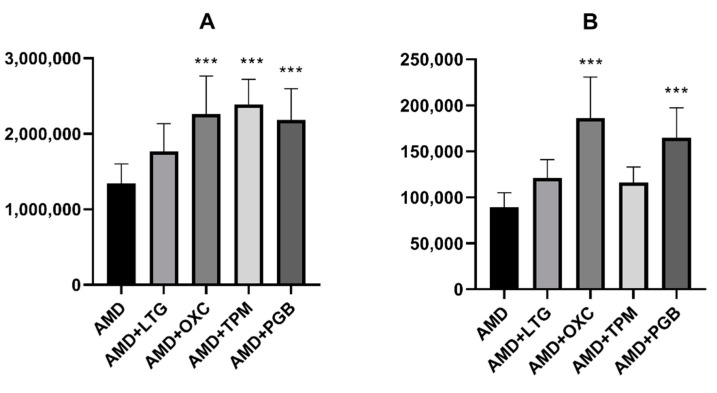
Effects of acute administration of antiepileptic drugs on the brain concentration of (**A**) amiodarone and (**B**) desethylamiodarone. Data are presented as the means ± SD of the peak area of at least eight determinations. Statistical analysis was performed using one-way analysis of variance (ANOVA) followed by the post hoc Dunett’s test. AMD—amiodarone, DEA—desethylamiodarone, LTG—lamotrigine, OXC—oxcarbazepine, TPM—topiramate, PGB—pregabalin, *** *p* < 0.001 vs. control group.

**Table 1 ijms-22-01041-t001:** Effects of acute amiodarone, antiepileptic drugs, and their combinations on motor performance in mice.

Drugs and Doses (mg/kg)	Animals Impaired (%)
Vehicle	0/10
AMD 100	0/10
LTG 6	0/10
LTG 6 +AMD 100	0/10
Statistics	-
Vehicle	0/10
AMD 100	0/10
OXC 8.9	0/10
OXC 8.9 +AMD 100	0/10
Statistics	-
Vehicle	0/10
AMD 100	0/10
TPM 53.7	1/10
TPM 53.7 + AMD 100	2/10
Statistics	*p*_(TPM)_ > 0.9999, *p*_(TPM + AMD)_ = 0.4737
Vehicle	0/10
AMD 100	0/10
PGB 57.8	8/10 ###
PGB 57.8 + AMD 100	6/10 #,●
Statistics	### *p* = 0.0007; #,● *p* = 0.0108

Data are expressed as percentage of animals that failed to perform the chimney test. Statistical analysis of data was calculated by using the Fisher’s exact probability test. AMD—amiodarone, LTG—lamotrigine, OXC—oxcarbazepine, TPM—topiramate; # *p* < 0,05 vs. Vehicle; ### *p* < 0,001 vs. Vehicle; ● *p* < 0,05 vs. AMD.

**Table 2 ijms-22-01041-t002:** Effects of acute amiodarone, antiepileptic drugs, and their combinations on long-term memory in mice.

Drugs and Doses (mg/kg)	Retention Time (s)
Vehicle	180 [176;180]
AMD 100	72 [25;100] #
LTG 6	180 [180;180]
LTG 6 +AMD 100	26 [18; 51] #,**
Statistics	KW = 20.21 *p* = 0.0002
Vehicle	180 [176;180]
AMD 100	72 [25;100] #
OXC 8.9	180 [180;180]
OXC 8.9 +AMD 100	48 [13; 108] ♦
Statistics	KW= 18.01 *p* = 0.0004
Vehicle	180 [176;180]
AMD 100	72 [25;100] ##
TPM 53.7	180 [180;180]
TPM 53.7 + AMD 100	79 [23;99] #,▲
Statistics	KW = 18.4 *p* = 0.0004
Vehicle	180 [176;180]
AMD 100	72 [25;100] #
PGB 57.8	144,5 [27;170]
PGB 57.8 + AMD 100	16 [13;23] ###
Statistics	KW= 19.34, *p* = 0.0002

Data are expressed as median retention time (with 25th and 75th percentiles) during which the animals avoided the dark compartment in the step-through passive-avoidance task. The results were analyzed using the nonparametric Kruskal–Wallis ANOVA test followed by Dunn’s post hoc test. AMD—amiodarone, LTG—lamotrigine, OXC—oxcarbazepine, TPM—topiramate; # *p* < 0,05 vs. Vehicle; ## *p* < 0,01 vs. Vehicle; ### *p* < 0,001 vs. Vehicle; ** *p* < 0,01 vs. LTG; ♦ *p* < 0,05 vs. OXC; ▲ *p* < 0,05 vs. TPM.

## Data Availability

The data presented in this study are available in the article.

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
