# Peer review of "Amiodarone Enhances Anticonvulsive Effect of Oxcarbazepine and Pregabalin in the Mouse Maximal Electroshock Model"

_ijms, 2021, doi:10.3390/ijms22031041_

Round 1

Reviewer 1 Report

This work is a continuation of Authors research on influence of different antiarrhythmic drugs on the activity of anticonvulsants. Previous work concerned amiodarone and its influence on anticonvulsant effect of carbamazepine in the mouse MES model. Presented paper deals with impact of amiodarone on the anticonvulsant activity of lamotrigine, oxcarbazepine, topiramate and pregabalin. ED50 values of the mentioned drugs were measured in connection with amiodarone. Motor coordination and long term memory effects in combination of amiodarone with anticonvulsants were estimated. Effects of amiodarone on  brain concentration of  antiepileptic drugs and antiepileptic drugs on the brain concentration of amiodarone and its metabolite desethylamiodarone were examined. It is interesting scientific contribution having practical meaning for the combined therapy of amiodarone and involved in the manuscript anticonvulsants. However it should be in the introduction justified the selection of antiepileptic drugs for the research. Each of these drugs has different structure and mechanism of anticonvulsant activity. Without of such reasoning it is difficult to perform well grounded discussion of the obtained effects. In my opinion the manuscript should be rewritten taking into the account suggested recommendation.

Reviewer 2 Report

This is an interesting manuscript. Attention to the following should improve it.
Minor comments:
1. In most journal publications, methods are discussed before results. If your format is not a journal requisite the order of the manuscript’s section be changed accordingly.
2. Although a change is not requisite, the extensive repetitive use of abbreviations makes the manuscript hard to read.
Major comments:
1. Table 1 is confusing. Consider using separate tables for the Chimney Test and the Step-through passive-avoidance test.
2. It is interesting that amiodarone had more influence on memory than coordination. In clinical practice ataxia is more common than confusion, disorientation, or delirium. This supports the notion that an analgesic effect may have influenced the results of the Step-through passive-avoidance test. You do a nice job of explaining this on page 8 of the discussion, but you may also wish to add that your findings at a higher amiodarone dose are consistent with the concept that the analgesic effect is dose dependent.
3. OXC and TPM induce CYP3a4 and might be expected to increase DEA as a result of amiodarone metabolism. However, it is a less potent inducer than CBZ and a significant pharmacokinetic interaction seems unlikely. PGB is not subject to hepatic metabolism and does not induce or inhibit liver enzymes such as the cytochrome P450 system. The main route of LTG metabolism does not involve cytochrome P450 enzymes. A pharmacokinetic explanation for increases in amiodarone alone or both amiodarone and DEA seems unlikely. It has been increasingly recognized that drug-drug interactions can affect the distribution of drugs into a particular compartment (e.g CNS) with or without affecting their systemic plasma (or blood) concentration. By modulating the blood brain barrier, a drug can affect the distribution of another drug into the brain, its removal from the brain, or both. Have you explored the potential influence of AEDs on amiodarone absorption through the blood brain barrier?

Reviewer 3 Report

Due to the growing numer of patients suffering both from seizures and heart rhythm disorders, studies detailing interactions between antiarrhythmic and anticonvulsive drugs attract increasing attention. This manuscript presents novel data regarding the enhancement of the anticonvulsive effects of oxcarbazepine and pregabalin by amiodarone in a mouse model of seizures.

In addition to the enhanced anticonvulsant action of oxcabazepine and pregabalin by amiodarone, key findings include impairment of long term memory function by amiodarone in mice in behavioral assessments. While the brain concentrations of all anticonvulsant drugs investigated in this study are not affected by amiodarone, oxcarbazepine and pregabalin (and topiramate, respectively) elevated the concentrations of amiodarone (and desethylamiodarone, respectively).

The study has been adequately carried out. The methods and the data are presented clearly and in a precise manner, the data is discussed extensively. However, due to the descriptive character of this study, a complete mechanistic image does not emerge from this study. As the authors mention in their discussion, it would be interesting to know more about the effects of chronic administration of the aforementioned drugs. Nevertheless, the results of this study might have important and relevant clinical implications for patients treated with a combination of amiodarone and anticonvulsant drugs.

Round 2

Reviewer 1 Report

After recommended changes article is suitable for publication in IJMS.

In the future it would be reasonable to  evaluate influence of examined drugs on P-glycoprotein activity (and grugs interaction)

Reviewer 2 Report

The authors have done a nice job of revising the manuscript.

Reviewer 3 Report

The manuscript can be accepted in its present form.